# Comparative Evaluation of the Dynamics of Animal Husbandry Air Pollutant Emissions Using an IoT Platform for Farms

Razvan Alexandru Popa [1,2], Dana Catalina Popa [1,*], Elena Narcisa Pogurschi [1], Livia Vidu [1], Monica Paula Marin [1], Minodora Tudorache [1], George Suciu [3], Mihaela Bălănescu [3], Sabina Burlacu [2,4], Radu Budulacu [2] and Alexandru Vulpe [2,4,*]

1   Formative Sciences in Animal Breeding and Food Industry Department, University of Agricultural Sciences and Veterinary Medicine, 59 Marasti Blv., 011464 Bucharest, Romania
2   R&D Department, Beam Innovation SRL, 041386 Bucharest, Romania
3   R&D Department, Beia Consult International, 041386 Bucharest, Romania
4   Telecommunications Department, University Politehnica of Bucharest, 061071 Bucharest, Romania
*   Correspondence: dana-catalina.popa@igpa.usamv.ro (D.C.P.); alex.vulpe@beaminnovation.ro (A.V.); Tel.: +40-720-210-178 (D.C.P.); +40-753-389-174 (A.V.)

**Abstract:** One of the major challenges of animal husbandry, in addition to those related to the economic situation and the current energy crisis, is the major contribution of this sector to atmospheric pollution. Awareness of pollution sources and their permanent monitoring in order to ensure efficient management of the farm, with the aim of reducing emissions, is a mandatory issue, both at the macro level of the economic sector and at the micro level, specifically at the level of each individual farm. In this context, the acquisition of consistent environmental data from the level of each farm will constitute a beneficial action both for the decision-making system of the farm and for the elaboration or adjustment of strategies at the national level. The current paper proposes a case study of air pollutants in a cattle farm for different seasons (winter and summer) and the correlation between their variation and microclimate parameters. A further comparison is made between values estimated using the EMEP (European Monitoring and Evaluation Programme, 2019) methodology for air pollutant emission and values measured by sensors in a hybrid decision support platform for farms. Results show that interactions between microclimate and pollutant emissions exist and they can provide a model for the farm's activities that the farmer can manage according to the results of the measurements.

**Keywords:** AP monitoring; IoT; AP estimation; decision support; livestock farming

## 1. Introduction

Currently, the exploitation of cows for milk production represents a major challenge, both from an economic point of view (the price of milk at the farm gate in relation to the current financial challenges) and in terms of its significant impact on the environment, i.e., the ecological implications at a global scale, noticed for several decades by the scientific community but widely and acutely felt in recent years.

The exploitation of animals for food production undoubtedly has a major ecological impact on the environment, represented in deforestation, crises in the provision of drinking water through the pollution of surface and underground waters, the loss of biodiversity at all its levels, emissions of greenhouse gases (approximately 30% of total emissions), pollutant emissions, etc. [1–4].

Ensuring food security is a priority policy of any modern state, but this goal must not be fulfilled under any circumstances. The concept of food security must also include environmental security in such a way that future generations are not affected.

Animal exploitation affects the environment on several levels. A first problem that arises is related to the pollution of water by increased levels of nutrients such as nitrogen and phosphorus, which causes the phenomenon of water eutrophication, with all its consequences

for the biodiversity of aquatic ecosystems. For this reason, the member states of the United Nations have included this phenomenon in the list of objectives "Sustainable Development Goals (SDGs) 14, 15, 17" (https://sdgs.un.org/goals, accessed on 18 October 2022).

Loss of specific biodiversity is another consequence of animal agriculture. The growing demand for animal products has led to an increase in cultivated areas. Under the conditions of the permanent expansion of urban areas, the provision of areas for the cultivation of fodder is achieved by deforestation. The FAO (Food an Agriculture Organization) has estimated that more than 7% of land area is used to feed dairy animals [5]. Deforestation leads, in addition to massive biodiversity loss [6], to the intensification of the greenhouse effect by releasing stored carbon in various forms.

A large number of gases that cause climate change are also known air pollutants that affect our health and the environment. In many ways, improving air quality can also boost climate change mitigation efforts and vice versa, but not always. The challenge is to ensure that climate change and air policies focus on win-win scenarios.

In its 2007 assessments, the Intergovernmental Panel on Climate Change (IPCC) predicted a decline in air quality in the future due to climate change, but we do not currently have a complete understanding of how climate change might affect air quality.

It is interesting to note that many climate-related processes are controlled not by the main components of our atmosphere, but by some gases that are found only in very small quantities. The most common of these so-called waste gases, carbon dioxide, makes up only 0.0391% of the air. Any variation in these very small amounts has the ability to affect and modify our climate.

The World Health Organization specifies that almost all the world's population (99%) breathes air with levels of pollutants that significantly exceed the maximum limits of admissibility and which, obviously, affect human health, generating different categories of pathologies (https://www.who.int/health-topics/air-pollution#tab=tab_1, accessed on 18 October 2022). These pollutants are represented by ammonia, volatile organic compounds, nitrogen oxides and microscopic particles in suspension. Animal production generates such pollutants; it is estimated that approximately 8% ($PM_{10}$) and 4% ($PM_{2.5}$) of the total microscopic particles in suspension come from this economic sector [7].

In this context and taking into account the fact that the demand for milk and dairy products is constantly increasing, one of the major challenges of milk production is represented by reducing its impact on the environment and minimizing pollutant emissions. Obviously, in this action we are limited by the physiology of the animal. As a result, the tools at our disposal are related to nutrition and farm management (type of feed, shelters, microclimate, feed administration, manure management).

The objectives of this study are:

- To present the current knowledge on the relationship between air pollutants and animal welfare and the relationship between air pollutant variation and farm management.
- To propose a case study in a real environment where an IoT infrastructure is used for monitoring key parameters of the stable environment: gas sensors ($NH_3$) and PM sensors ($PM_{2.5}$, $PM_1$, $PM_{10}$).
- To estimate the air pollutants' concentrations (in two seasons—winter and summer) based on European Monitoring and Evaluation Programme (EMEP) methodology and to compare the estimated values with the monitored concentrations.
- To study the behavior of air pollutants in correlation with micro-climate parameters.

The paper is organized as follows. Section 2 reviews the state of the art in assessing the relationship between air pollutants, animal welfare and farm management. Section 3 describes the architecture of the platform for AP monitoring and outlines the case study conducted using the platform. Sections 4 and 5 present the results obtained and the conclusions drawn, respectively.

## 2. State of the Art

Pollutant emissions from animal husbandry must benefit from a holistic treatment, as they influence each other. In order to understand the phenomenon, it is necessary that the physiology of cows be introduced in the context of trophic relations in a biocenosis. Thus, it must be accepted that the Eltonian pyramid is a simplistic representation of trophic relationships, and the very strict labeling of a trophic link is a gross error. The stability of a biocenosis by optimizing energy and nutrient flows would not be possible without some plasticity in the component species. Thus, cows can be considered the most inefficient herbivores, trophically speaking, and, to use a figure of speech, "the most carnivorous of herbivores." For this reason, cows are the most inefficient organisms regarding nitrogen utilization. It has been found that between 50 and 80% of consumed nitrogen is excreted as urea and other nitrogen compounds through feces and urine [8] (a consequence of physiology and trophic position), which represent important sources of ammonia emissions [9]. Most of the nitrogen is present in the urine and a smaller part in the feces. Regardless of the operating conditions, in the stable or on the pasture, at the time of excretion and the mixture of urine and feces, the nitrogen in the form of urea in the urine is converted into an unstable mixture of ammonia ($NH_3$) and ammonium ($NH_4$) under the action of urease from feces, resulting ammonia volatilization [10].

Numerous research papers have highlighted the direct relationship between the nitrogen content (in various forms) of cow manure and their nutrition. The crude protein content of the ration influences ammonia emissions through the manure, even establishing a linear relationship between them [11]. Decreasing the crude protein content of the ration is proving to be an effective means of reducing ammonia emissions from dairy cow manure [12], with research showing a 45% reduction in ammonia emissions as a result of lowering from 17% to 13.5% of crude protein content in cow diets [13]. Swensson [14] finds that ammonia emissions will be three times higher when the crude protein content of the ration increases from 13 to 19%. Similar results, namely the highlighting of the direct relationship between crude protein intake and ammonia emissions, are reported by other authors [15–18].

It should be emphasized that the efficiency of nitrogen use in the metabolism of dairy cows is also influenced by other nutritional factors apart from crude protein intake. For example, in cows, the nitrogen content of milk, under the conditions of a diet with similar levels of protein, depends on the content and composition of carbohydrates in the ration [19]. The nitrogen content of milk will decrease (so we expect higher values in urine and feces) on a high-fiber diet compared to a high-starch diet.

However, it should be kept in mind that ammonia emissions from manure (urine) also depend on other factors. Research has shown that the type of soil, atmospheric humidity, temperature, wind speed or air currents in the shelter influence this aspect and cause large variations in ammonia losses from urine: between 25–50% [9] or between 4–52% [20]. In this regard, significant results were achieved [21] stating that ammonia emissions increase with temperature, and that this is directly related to the type of floor in the shelter and the management of manure. Significant differences occur depending on the cow maintenance system. In open systems the manure will be deposited directly on the soil and there will thus be a rapid conversion of urea to ammonia (i.e., high emissions), while in closed systems, with the maintenance of cows in the shelter, the emissions are lower as a result of the regular removal of manure from the shelter [22,23]. Inside the stable, ammonia emissions differ significantly depending on the type of flooring. Solid floors generate higher emissions than those in the form of a grid [24], because the former facilitate the mixing of feces and urine. Numerous research papers have shown that the temperature inside the stable is also an important factor in ammonia emission, its high values causing higher ammonia emissions (through its relation to urease activity), which is also generating seasonal differences [25,26], especially in temperate climates.

Microscopic particles are particles suspended in the air and produced in various types of industries and agriculture. A high concentration of them affects human and animal health. In animal husbandry in general, and in cow farms in particular, the sources

of air pollution with microscopic particles could be feed administration [27] and feed management (wet vs. dry food, feed distribution system, feed storage), waste burning [28], stable cleaning, manure management, animal movement, animal maintenance (on bedding or not), ventilation rate or the microclimate in the shelter, while an indirect origin is oxidation of ammonia or other precursor gases [29–31].

Obviously, in agriculture, as in other types of industries, the degree of pollution with microscopic particles both at the global level and at the point level (stable, farm) depends on a number of factors: climatic zone, season, geographical peculiarities, humidity etc. [32]. The limitation of air pollution with microscopic particles belongs exclusively to the management of the farm, given the fact that the other influencing factors cannot be controlled, but only possibly influenced to a small extent.

One possible way of limiting particle emissions is by carrying out certain farm operations (particle generators) during the night. This is based on the fact that during the day the concentration of particles can increase 10–15 times [33] as a result of the simultaneous action of four factors: moisture at the soil surface, air humidity, the angle of the sun's rays and temperature.

Manure management can result in a decrease in farm-level particulate emissions. Removing litter and storing it in non-concentrated forms can be a useful action, along with maintaining a layer of 2–3 cm of concentrated manure mixed with soil [34].

Other research has highlighted that reducing particle pollution in cow farms can be possible through the use of a sprinkler system [35], but this solution is debatable under the current water crisis.

The reduction of atmospheric pollutant emissions must represent a priority of each economic sector, and the lack or inefficiency of technical and/or legal mechanisms for their permanent monitoring attracts behaviors that are not related to ensuring sustainability. Obviously, these mechanisms must be subject to a general legal framework, but the awareness of each farmer of his contribution to the total amount of emissions at the national level is a matter of common sense and absolutely mandatory. In this sense, the implementation of technical solutions for monitoring emissions at point level (at the farm level) and for alerting in case of exceeding them would allow farmers to achieve efficient management and create a particularly useful national system that would enable strategy correction in real time.

The growing urbanization trend has led to a great deal of research conducted on the modification of air pollutant emissions due to climate modification. For example, a study has shown [36] that air quality deterioration may be caused by anthropogenic activities and land use changes. Air pollutants such as VOC, $O_3$, PMs or NOx are found to be correlated with density and population, but also with mean summer temperature and precipitation.

In addition, a considerable number of studies show that micro-climatic features have an impact on air quality and thermal comfort. For example, in Sri Lanka [37] indoor concentrations of $CO_2$, $NO_2$, $PM_{2.5}$, CO, VOC, temperature, relative humidity and wind speeds were measured. The findings of the study recommended the introduction of a vegetative cover around buildings in suburban areas to overcome this problem since vegetation has a favorable impact on temperature and the concentrations of several air pollutants.

## 3. Measurement Platform for AP Concentration Monitoring and Case Study Description

### 3.1. Platform Architecture

AP concentrations in a case study are obtained through two methods: (i) estimation based on EMEP/EEA methodologies and (ii) monitoring using various sensors (IoT devices). Based on the EMEP/EEA equations, an IoT-based hardware–software platform was designed, tested and implemented. The information gathered by this system (concerning stable environment, AP concentration and animal state) and the way that information is processed, stored and presented allow it to be used with ease by farmers.

The platform has been presented in detail in a previous paper [38]; Figure 1 illustrates the platform architecture.

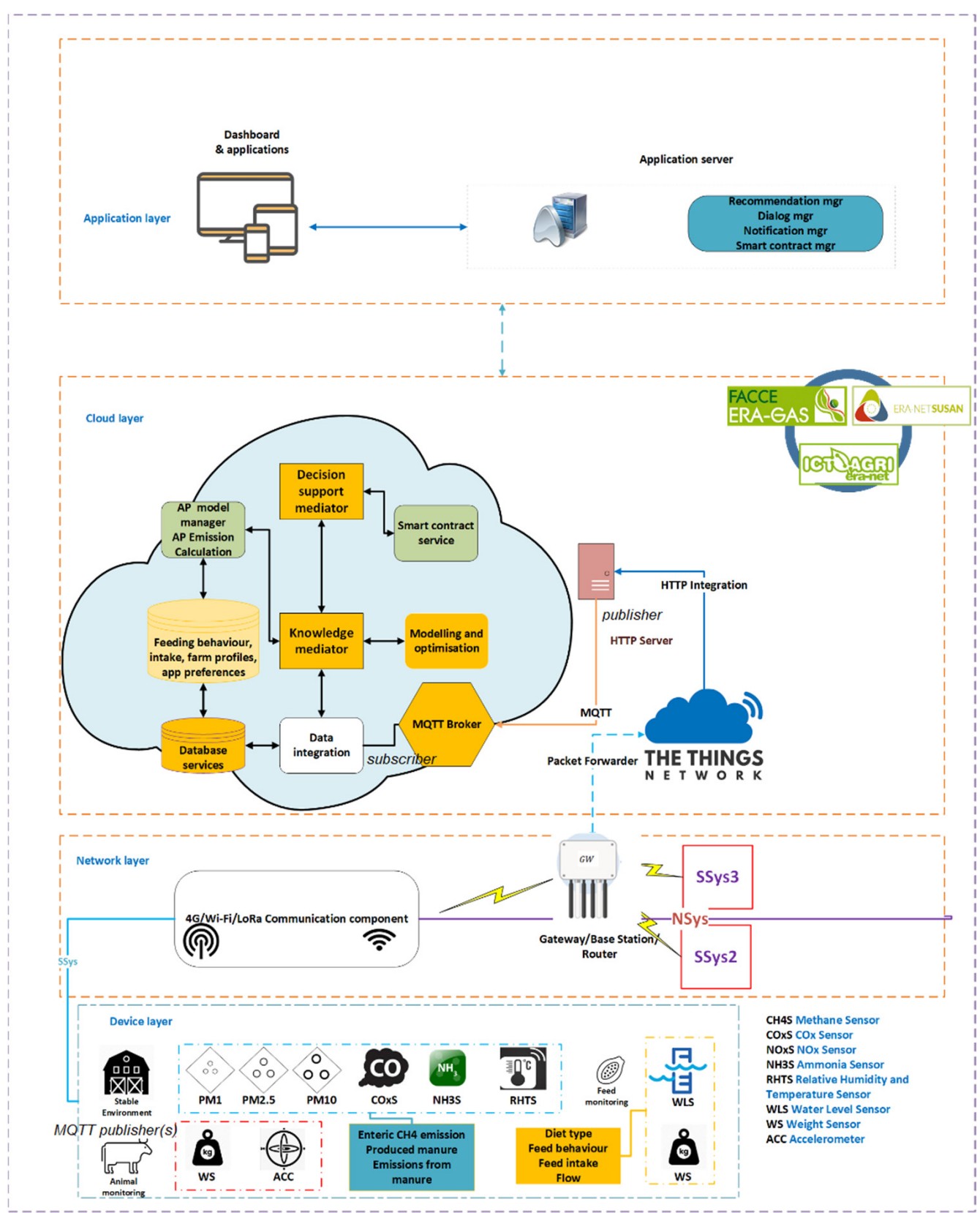

**Figure 1.** IoT-based AP Monitoring platform architecture.

The high-level architecture of the proposed IoT platform is divided into four layers:

- Device Layer: includes sensors (i.e., for measurement of the concentrations of APs such as $NH_3$, CO, CO and PMx), devices and client agents (to collect and transmit data to the IoT platform).
- Network Layer: includes the communication component (which uses low-power radio transmission technologies such as LoRa and cellular IoT) and the gateway (which sends the data packets to the next Layer).
- Cloud Layer: has the role of transforming data into knowledge. In this way, intelligence is added as a higher level of services. This layer receives the data and integrates and transforms them into knowledge. The data are received through the use of The Things Network and MQTT protocol.
- Application Layer: uses the knowledge generated in the previous layer to provide an overview of the farm performance (based on specific KPIs such as productivity, AP concentrations etc.) and their visual representation (various graphic representations).

Data is collected via diverse sensors which are equipped with different wireless interfaces: either 4G, Wi-Fi or LoRa. They transmit data packets via a device that allows connection to the Internet, called Base Station (for 4G), Router (for Wi-Fi) or Gateway (for LoRa). The data packets are transmitted via MQTT protocol either directly (for 4G and Wi-Fi) or via the Things Network (for LoRa). Finally an MQTT broker receives the data which is at this point being used by the Cloud Layer.

The platform presented uses open-source software and offers the possibility of long-term operation (>10 years) for IoT devices within the platform without battery replacement. We argue that the platform is sustainable (i) from an environmental point of view and (ii) in terms of the human resources needed for the operation and future extension of the platform.

### 3.2. Case Study

The input data for the estimation of atmospheric pollutant emissions are provided from a dairy cow farm, which has in operation (intensive system, no pasture or paddock) 120 dairy cows, 40 heifers and primiparous and 40 youth heads (3–9 months). The animals are from the Montbeliarde and Friza breeds. The stable is cleaned twice a day, and the feeding is done with the technological trailer used for feed distribution. The manure is separated into a solid fraction (deposited on the solid storage platform) and a liquid fraction (in the sealed lagoon).

The monitoring of the farm environment is done using the IoT infrastructure listed on Table 1 which transmits data wirelessly according to the architecture in Figure 1. These on-farm wireless sensors transmit information needed to determine the source of AP emissions and the costs associated with these emissions. Figure 2 shows that the IoT devices are carefully positioned in locations where they do not interfere with the daily activities of the animals.

**Table 1.** Sensors used in the IoT infrastructure.

| Sensor Name | Parameter | Measurement Unit | Minimum Measured Value | Maximum Measured Value |
|---|---|---|---|---|
| BME280 [1] | Temperature | °C | 0 | 65 |
| BME280 | Humidity | % RH | 0 | 100 |
| BME280 | Pressure | kPa | 30 | 110 |
| MICS-6814 [2] | CO | ppm | 30 | 1000 |
| MICS-6814 | $NO_2$ | ppm | 0.05 | 5 |
| CCS_811 [3] | $CO_2$ | ppm | 350 | 10,000 |
| SEN0237-A [4] | $O_2$ | % | 0 | 30 |
| CCS811 | VOC | ppm | 30 | 400 |
| OPC-N2 [5] | PM1 | $\mu g/m^3$ | 0 | 1 |
| OPC-N2 | PM2.5 | $\mu g/m^3$ | 0 | 2.5 |
| OPC-N2 | PM10 | $\mu g/m^3$ | 0 | 10 |

[1] https://www.bosch-sensortec.com/products/environmental-sensors/humidity-sensors-bme280/, accessed on 18 October 2022; [2] https://www.sgxsensortech.com/content/uploads/2015/02/1143_Datasheet-MiCS-6814-rev-8.pdf, accessed on 18 October 2022. [3] https://learn.adafruit.com/adafruit-ccs811-air-quality-sensor, accessed on 18 October 2022. [4] https://ro.farnell.com/dfrobot/sen0237-a/analog-dissolved-oxy-sensor-kit/dp/3517931, accessed on 18 October 2022. [5] http://www.aqmd.gov/aq-spec/product/alphasense, accessed on 18 October 2022.

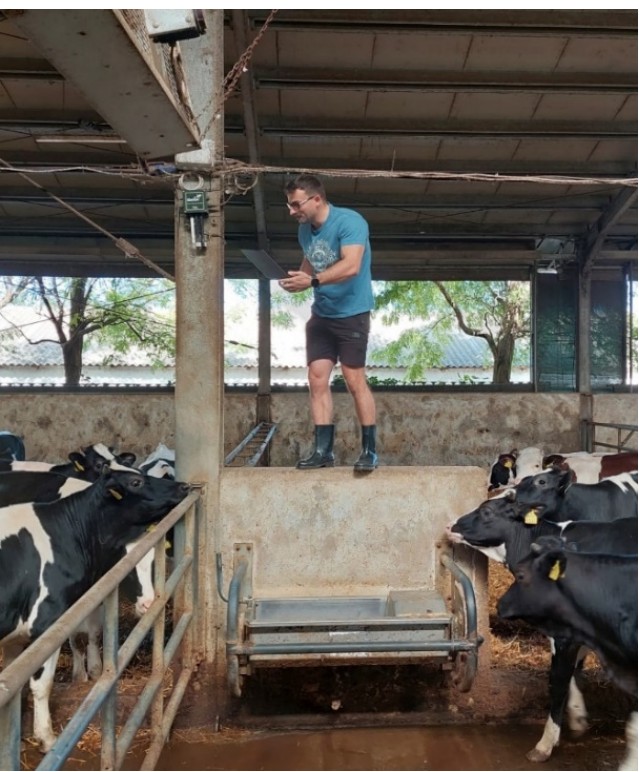

**Figure 2.** IoT devices installed in the farm and their configuration.

## 4. Comparison between Estimated and Monitored AP Concentrations

The usefulness of a support platform for monitoring environmental factors and pollutant emissions lies in two aspects: it is necessary on the one hand from the perspective of environmental protection (for monitoring emissions), and on the other hand from the perspective of farm management (as a support system in the decision-making process through emission monitoring and alert systems).

### 4.1. AP Concentration Estimated Using EMEP Methodology

This section will present the estimation of air pollutant (AP) concentrations using the EMEP methodology.

The EMEP 2019 guideline (the co-operative programme for monitoring and evaluation of the long-range transmission of air pollutants in Europe—unofficially European Monitoring and Evaluation Programme) proposes different methods of estimating the emissions of pollutants from animal husbandry, depending on how many parameters are known in each case. For example, Tier 1 is the simplest method for estimating the pollutant emission as it consists of the multiplication of the default emission factor by the number of animals specific to each category (this method was used for PM emissions estimation).

The next level, if more parameters are known, can use the tier 2 method, which involves specific parameters. In our study, we used tier 2 for estimating $NH_3$ emissions (EMEP/EEA guideline, 2019) and we combined this value with tier 2 for the calculation of excreted nitrogen from the IPCC (Intergovernmental Panel on Climate Change) 2019 guideline [39].

Methane emissions were estimated using the manure management N-flow tool (https://www.eea.europa.eu/publications/emep-eea-guidebook-2019/part-b-sectoral-guidance-chapters/4-agriculture/manure-management-n-flow-tool, accessed on 18 October 2022).

There are no specific equations in the IPCC or EMEP guidelines for CO and in our case study the data refers exclusively to the sensors' measurements. Table 2 presents the parameters and equations used for AP emissions.

**Table 2.** Parameters used for the estimation of AP emissions and methodology.

| Crt. No. | Parameter | Guideline | Equation/Table Number in IPCC, 2019 and EMEP, 2019 |
|---|---|---|---|
| | | Calculated parameters | |
| 1 | $N_{ex}$ | IPCC, 2019 | 10.31 A |
| 2 | $N_{intake}$ | IPCC, 2019 | 10.32 |
| 3 | $N_{retention}$ | IPCC, 2019 | 10.33 |
| 4 | $NE_g$ | IPCC, 2019 | 10.6 |
| 5 | $m_{hous\_N}$ | EMEP, 2019 | 5 |
| 6 | $m_{hous\_TAN}$ | EMEP, 2019 | 10 |
| 7 | $m_{hous\_solid\_N}$ | EMEP, 2019 | 14 |
| 8 | $E_{hous\_solid}$ | EMEP, 2019 | 16 |
| 9 | $E_{storage\_solid}$ | EMEP, 2019 | 34 |
| 10 | $E_{MMS\_NH3}$ | EMEP, 2019 | 46 |
| | | Default values | |
| 1 | $X_{TAN}$ | EMEP, 2019 | Table 3.9 |
| 2 | $EF_{housing}$ | EMEP, 2019 | Table 3.9 |
| 3 | $EF_{PM2.5}$, $EF_{PM10}$ | EMEP, 2019 | Table 3.5 |

The primary data needed to calculate excreted nitrogen ($N_{ex}$) (equations 10.32 and 10.33—see Table 2) are provided by the farm. The rations administered to all three categories fluctuate during the year, but the differences between the summer and winter seasons are insignificant. Depending on the animal's age and the category of exploitation the rations were calculated using the structure and chemical composition of the feed in each case. Feeding dairy cattle is very important for ensuring constant and qualitative production throughout the year. Romanian farmers with an important herd of dairy cows choose the option of establishing a unique forage recipe, so that the animals on the farm benefit from the same food all year round. The uniformity of the feed contributes to ensuring a uniform amount of milk and does not allow farmers to change the qualitative parameters of the milk.

In our case study, during the summer season dairy cows, primiparous and heifers receive the green mass (5 kg maximum, introduced gradually, because the rumen is not accustomed to this type of feed after the winter season), and in the cold season for these categories the green mass is replaced with beer mug, without important changes in the value of the ingested energy. Between the summer and winter seasons there exist differences regarding $N_{ex}$ values (dairy cattle, primiparous and heifers). For the young (3–9 months), a single ration is provided throughout the year, and for primiparous and heifers there are no differences between seasonal $N_{ex}$ values.

Table 3 presents the feed structure of rations for both seasons and for all animal categories.

**Table 3.** Fodder type used in all cattle categories and chemical structure (per kilo).

| Category | PB (g/kg) | GB (g/kg) | CelB (g/kg) | SEN (g/kg) | GE (Kcal/kg) |
|---|---|---|---|---|---|
| Barley straw | 32.00 | 14.00 | 390.00 | 381.00 | 3772.91 |
| Alfalfa hay | 120.00 | 30.00 | 330.00 | 340.00 | 3969.90 |
| Corn silage | 22.00 | 8.00 | 85.00 | 127.00 | 1138.58 |
| Corn kernels | 90.00 | 40.00 | 22.00 | 710.00 | 3960.88 |
| Barley kernels | 100.00 | 20.00 | 56.00 | 678.00 | 3857.50 |
| Rape seed meal | 350.00 | 25.00 | 130.00 | 312.00 | 4163.24 |
| Wheat bran | 150.00 | 40.00 | 105.00 | 530.00 | 3951.05 |
| Soybean meal | 443.00 | 14.00 | 63.00 | 311.00 | 4265.60 |
| Beer wort | 50.40 | 15.20 | 38.10 | 70.00 | 907.09 |
| Green mass | 31.00 | 4.80 | 60.00 | 70.00 | 802.22 |

where, PB = crude proteine; GB = crude fat; CelB = crude cellulose; SEN = non-nitrogenous extractable substances; GE = gross energy intake.

The feed categories presented in Table 3 are cut to the required dimensions, mixed and homogenized, and are presented in the form of a unique mixture, balanced according to the physiological and production requirements of the animals.

For all categories, the unique mixture is administered throughout the year with the technological trailer. In summer green mass is added, and in winter beer wort is incorporated into the mixture. Table 4 presents the composition of that mixture for dairy cattle, primiparous and heifers during the year (summer period and winter period).

**Table 4.** Composition of unique mixture.

| Category | Animal Category | | | | | |
| | Dairy Cattle | | Primiparous and Heifers | | Youth (3–9 Months) | |
| | Summer | Winter | Summer | Winter | Summer | Winter |
| | kg/Head/Day | kg/Head/Day | kg/Head/Day | kg/Head/Day | kg/Head/Day | kg/Head/Day |
|---|---|---|---|---|---|---|
| Barley straw | 0.5 | 0.5 | 1.5 | 1.5 | 1.0 | 1.0 |
| Alfalfa hay | 2.0 | 2.0 | 3.5 | 3.5 | 2.0 | 2.0 |
| Corn silage | 20.0 | 20.0 | 15.0 | 17.0 | 9.0 | 9.0 |
| Corn kernels | 3.5 | 3.8 | 3.5 | 4.0 | 2.5 | 2.5 |
| Barley kernels | 1.5 | 1.5 | 1.5 | 1.0 | 0.3 | 0.3 |
| Rape seed meal | 2.5 | 2.5 | 2.0 | 2.0 | 0.3 | 0.3 |
| Wheat bran | 2.0 | 2.0 | - | - | - | - |
| Soybean meal | 2.0 | 2.0 | 2.0 | 2.0 | - | - |
| Beer wort | - | 8.0 | - | 3.0 | - | - |
| Green mass | 8.0 | - | 5.0 | - | - | - |
| Total | 42.0 | 42.3 | 34.0 | 34 | 15.1 | 15.1 |

A total mixed ration (TMR) was administered consisting of alfalfa hay, barley straw, corn silage, corn and barley grain colza meal, soya meal, the proportion of feeds varying according to the need for nutrients in the categories of tested cattle.

During the summer the TMR was supplemented with green mass represented by alfalfa, which is a food rich in the nutrients necessary for production. During the winter the TMR was supplemented with by-products represented by beer wort.

The rations are balanced from the point of view of both macro- and micro-nutrients, which allows farmers to obtain increased milk production, a development of pregnancy in optimal conditions and an average daily gain corresponding to the young cattle.

For the formulation and optimization of the rations administered to the Holstein taurine categories, a list of fodder was established that included fibrous, coarse and concentrated fodder. This feed structure of the rations has been optimized so as to cover the entire nutrient requirement for dairy cows, pregnant cows and heifers and young cattle.

In the composition and optimization of the rations, fodder and by-products of plant origin were introduced in order to efficiently use the local fodder resources and with an optimal cost price.

To calculate the caloricity of the gross energy intake of each recipe or ration, the following equivalences were considered [35] (p. 114):

$$1 \text{ g crude protein} = 5.72 \text{ kcal};$$

$$1 \text{ g crude fat} = 9.5 \text{ kcal};$$

$$1 \text{ g crude fibers} = 4.79 \text{ kcal};$$

$$1 \text{ g SEN (non-nitrate extractable substances)} = 4.17 \text{ kcal}.$$

The GE calculation formula [35] is (p. 131):

$$GE \text{ (kcal/kg)} = 5.72 \cdot GP + 9.5 \cdot GB + 4.79 \cdot CelB + 4.17 \cdot SEN$$

where:

GE = gross energy intake
GP = crude protein
GB = crude fat
CelB = crude fibers
SEN = non-nitrate extractable substances

The rations were calculated according to this equation, and the values of crude protein, crude fat, crude fibers and non-nitrate extractable substances were taken from the tables with the feed chemical composition [35] (pp. 513–517).

In accordance with the requirements of the IPCC 2019 that the energy be expressed in MJ/kg in the calculation of ratios, we multiplied the values by 10 to express the caloricity for 1 kg (tables give the value of these nutrients expressed as a percentage, for 100 g).

The total value of the ration, expressed in kcal, was divided by 239 in order to obtain the equivalence in MJ (Mega Joules).

The equivalence relations are as follows [35] (p. 114):

$$1 \text{ MJ} = 239 \text{ kcal}$$

where MJ = megajoule and Kcal = kilocalory

For each feed category, the values of crude protein, crude fat, crude fibers and non-nitrate extractable substances are included in a table [35] (pp. 513–517); these table values are multiplied by the caloricity specific to each nutrient (5.72 kcal for 1 g of crude protein, etc.), followed by the adding of the caloricity of each nutrient and the achievement of the respective forage caloricity. This value is multiplied by the number of feed kilograms specified in the ration.

Table 5 presents total gross energy (GE) expressed in MJ per head and per day, for each category of animals and for both seasons.

**Table 5.** Total gross energy intake (MJ/head/day).

| Category Season | Dairy Cows | Heifers and Primiparous | Youth (3–9 Months) |
|---|---|---|---|
| Winter | 330.91 | 270.33 | 143.40 |
| Summer | 335.90 | 280.10 | 143.40 |

Table 6 presents the specific parameters used for calculation of ammonia emissions.

**Table 6.** Parameters' values used for calculated excreted nitrogen ($N_{ex}$).

| Category Parameter | Dairy Cows | | Heifers and Primiparous | | Youth (3–9 Months) |
|---|---|---|---|---|---|
| Days of life | 365 | | 365 | | 180 |
| Heads number | 120 | | 40 | | 40 |
| AAP | 365 | | 365 | | 19.73 |
| Season | Summer | Winter | Summer | Winter | All year |
| GE (MJ/head/day) | 330.91 | 335.90 | 270.33 | 280.10 | 143.40 |
| CP% (%) | 0.143 | 0.160 | 0.161 | 0.170 | 0.205 |
| Milk (kg/head/day) | 30 | 28 | - | - | - |
| Milk% (%) | 1.92 | 1.92 | - | - | - |
| WG (kg/day) | 0.2 | 0.2 | 0.4 | 0.4 | 0.9 |
| $NE_g$ (MJ/head/day) | 1.96 | 1.96 | 3.93 | 8.31 | 6.05 |
| $N_{ex}$ (kg/head/year) | 119.83 | 139.78 | 135.06 | 147.55 | 81.40 |

where: AAP = average annual population = number of animals produced annually × days of live. GE = gross energy intake (MJ/head/day). CP% = percent crude protein in dry matter (%). Milk = milk production (kg/head/day). Milk% = percent of protein in milk, calculated as [1.9 + 0.4 × %Fat], where %Fat was determined by milk analyzer (Farm Eco 25) = 4% (%). WG = weight gain (kg/day). $NE_g$ = net energy for growth, calculated in livestock characterization, based on current weight, mature weight, rate of weight gain, and IPCC constants (MJ/head/day). $N_{intake}$ = daily N consumed per animal of each category (kg N/head/day). $N_{retention}$ = amount of daily N intake by head of animal (kg N/head/day). $N_{ex}$ = annual N excretion rates (kg N/head/year).

The percentage of crude protein in dry matter (CP%) was calculated based on the chemical composition of each fodder and then multiplied by the proportion of feed in the total ratio.

Table 7 presents the estimated $NH_3$ emissions.

**Table 7.** Estimated $NH_3$ emissions during the monitoring period.

| Animal Category | NH₃ (t/Year) | | |
| --- | --- | --- | --- |
| | **Season** | | |
| | **Summer (185 Days)** | **Winter (180 Days)** | **All Year (365 Days)** |
| *Dairy cattle* | 1.21 | 1.40 | 2.61 |
| *Primiparous and heifers* | 0.48 | 0.5 | 0.98 |
| *Young (3–9 months)* | 0.92 | | 0.92 |
| **Total** | | **4.51** | |

Table 8 presents the estimated microscopic particles emissions.

**Table 8.** Microscopic particles emissions ($PM_{2.5}$, $PM_{10}$) (kg PM/year).

| Category | Heads No | Life Days | AAP | EF | | Emissions (kg/Year) | |
| --- | --- | --- | --- | --- | --- | --- | --- |
| | | | | **PM₁₀** | **PM₂.₅** | **PM₁₀** | **PM₂.₅** |
| Dairy cows | 120 | 365 | 110 | 0.63 | 0.41 | 75.6 | 49.2 |
| Heifers and primiparous | 40 | 365 | 50 | 0.63 | 0.41 | 25.2 | 16.4 |
| Young | 40 | 180 | 19.73 | 0.27 | 0.18 | 10.8 | 7.2 |
| **Total** | | | | | | **111.6** | **72.8** |

### 4.2. AP Concentration Monitored Using Sensors

The validation of an interactive platform model that can be used for farm management from the perspective of microclimate parameters and pollutant emissions must be based on the study of their behavior, alone and in relation to one another. The complexity of the relationships between them, of a physical-chemical nature, makes taking a parameter out of context and studying its behavior, abstracting it from its interaction with others, a dead end.

In this sense, using the data obtained from two studied seasons (summer and winter) regarding the microclimate parameter values (humidity and temperature) and the studied pollutant emission values, respectively **711** records in the winter season and **597** records in the summer season, the correlations between them and their meaning and significance were determined. Tables 9 and 10 present the correlation values between the studied parameters and their significance over two measurement seasons.

**Table 9.** Correlation between microclimate and pollutant concentrations during the winter season (711 recordings).

| Specification | HUM | NH₃ | PM₁ | PM₁₀ | PM₂.₅ | T⁰C |
| --- | --- | --- | --- | --- | --- | --- |
| HUM | - | 0.56$^{HS}$ t = 17.99 | 0.43$^{HS}$ t = 12.68 | 0.02$^{NS}$ t = 0.53 | 0.24$^{HS}$ t = 6.58 | -0.67$^{HS}$ t = 24.03 |
| NH₃ | | - | 0.41$^{HS}$ t = 11.97 | 0.12$^{S}$ t = 3.21 | 0.33$^{HS}$ t = 9.31 | -0.02$^{NS}$ t = 0.53 |
| PM₁ | | | - | 0.13$^{S}$ t = 3.49 | 0.71$^{HS}$ t = 26.84 | -0.23$^{HS}$ t = 6.29 |
| PM₁₀ | | | | - | 0.42$^{HS}$ t = 12.32 | 0.06$^{NS}$ t = 1.60 |
| PM₂.₅ | | | | | - | -0.02$^{NS}$ t = 0.53 |

HUM = humidity (%). t = correlation significance. S = significant, $p < 0.05$. S = significant, $p < 0.01$. HS = high significant, $p < 0.001$. NS = nonsignificant, $p > 0.05$. Critical $t_{0.05}$ = 1.96. Critical $t_{0.01}$ = 2.58. Critical $t_{0.001}$ = 3.29.

**Table 10.** Correlation between microclimate and pollutant concentrations during the summer season (597 recordings).

| Specification | HUM | NH$_3$ | PM$_1$ | PM$_{10}$ | PM$_{2.5}$ | T$^0$C |
|---|---|---|---|---|---|---|
| HUM | - | 0.25$^{HS}$<br>t = 6.30 | 0.18$^{HS}$<br>t = 4.46 | −0.31$^{HS}$<br>t = 7.95 | −0.18$^{HS}$<br>t = 4.46 | −0.89$^{HS}$<br>t = 47.61 |
| NH$_3$ | | - | 0.11$^{S}$<br>t = 2.70 | −0.03$^{NS}$<br>t = 0.73 | 0.002$^{NS}$<br>t = 0.048 | −0.25$^{HS}$<br>t = 6.29 |
| PM$_1$ | | | - | 0.39$^{HS}$<br>t = 10.33 | 0.65$^{HS}$<br>t = 20.86 | −0.01$^{NS}$<br>t = 0.24 |
| PM$_{10}$ | | | | - | 0.80$^{HS}$<br>t = 32.52 | 0.32$^{HS}$<br>t = 8.23 |
| PM$_{2.5}$ | | | | | - | 0.25$^{HS}$<br>t = 6.30 |

HUM = humidity (%). t = correlation significance. S = significant, $p < 0.05$. S = significant, $p < 0.01$. HS = high significant, $p < 0.001$. NS = nonsignificant, $p > 0.05$ Critical t$_{0.05}$ = 1.96. Critical t$_{0.01}$ = 2.58. Critical t$_{0.001}$ = 3.29.

From the analysis of the results presented in Tables 9 and 10 we can observe the existence of correlations with varying degrees of significance between the values of the microclimate parameters and the pollutant emissions from the stable, different between the two seasons. This cumulative behavior raises certain problems related to the management of the farm. Thus, in the winter season, we can observe that the humidity in the animal stable correlates intensely negatively with the temperature, the increase of one implicitly leading to the decrease of the other. Furthermore, the increase in humidity is not related to PM$_{10}$ emissions, the latter being exclusively related to farm management, especially feed administration, stable cleaning actions and other possible occurrences. As a result, the increase in PM$_{10}$ emissions cannot be attributed to the existence of vapor supersaturation of the air, and the eventual switching on of fans to reduce humidity would do nothing more than circulate microscopic particles through the air. Conversely, increased humidity will maintain in the air a high concentration of the other types of particulate matter (PM$_1$ and PM$_{2.5}$) as well as ammonia. In the case of ammonia, when the temperature increases, the proportion of stable emissions becomes lower. Regarding particulate matter, it is noted that its concentration in the air is related exclusively to the concentration of ammonia, and not at all to the temperature.

In the warm season, increased humidity will lead to a decrease in the concentration of PM$_{10}$ and PM$_{2.5}$. In the summer, it seems that the concentrations of particulate matter in the air (PM$_{10}$ and PM$_{2.5}$) do not correlate with those of ammonia, and consequently the reduction of particle emissions must be attributed to activities related to food administration, cleaning, the evacuation of solid waste, and the reduction of ammonia by good ventilation. But the significantly positive correlation between PM$_{10}$ and temperature indicates that, during the warm season, the reduction of PM$_{10}$ emissions is done by lowering the temperature, that is, by ventilation and possibly by air humidification systems (sprinklers).

A clearer picture of the interactions between microclimate and pollutant emissions is presented in Figures 3 and 4 for the two seasons, showing the correlations between the observations by trendline using zero-intercept linear regression.

Analysis of the scatterplots presented in Figures 3 and 4 graphically reveals the links between the analyzed variables. Thus, the shape of the cloud of points, the direction of the regression line and the value of R$^2$ emphasize the strength of the links. Strongly correlated values are highlighted by the existence of a cloud of points aligned along the regression line. The interruption of the cloud of points (Figure 3m) suggests the existence, with a preponderance, of extreme values of PM$_{10}$ and PM$_{2.5}$ in the winter season. Points that lie outside the cloud suggest either outliers (possible registration errors) or drastic increases in a certain parameter. Obviously, the direction of the regression line suggests the algebraic sign of the correlation.



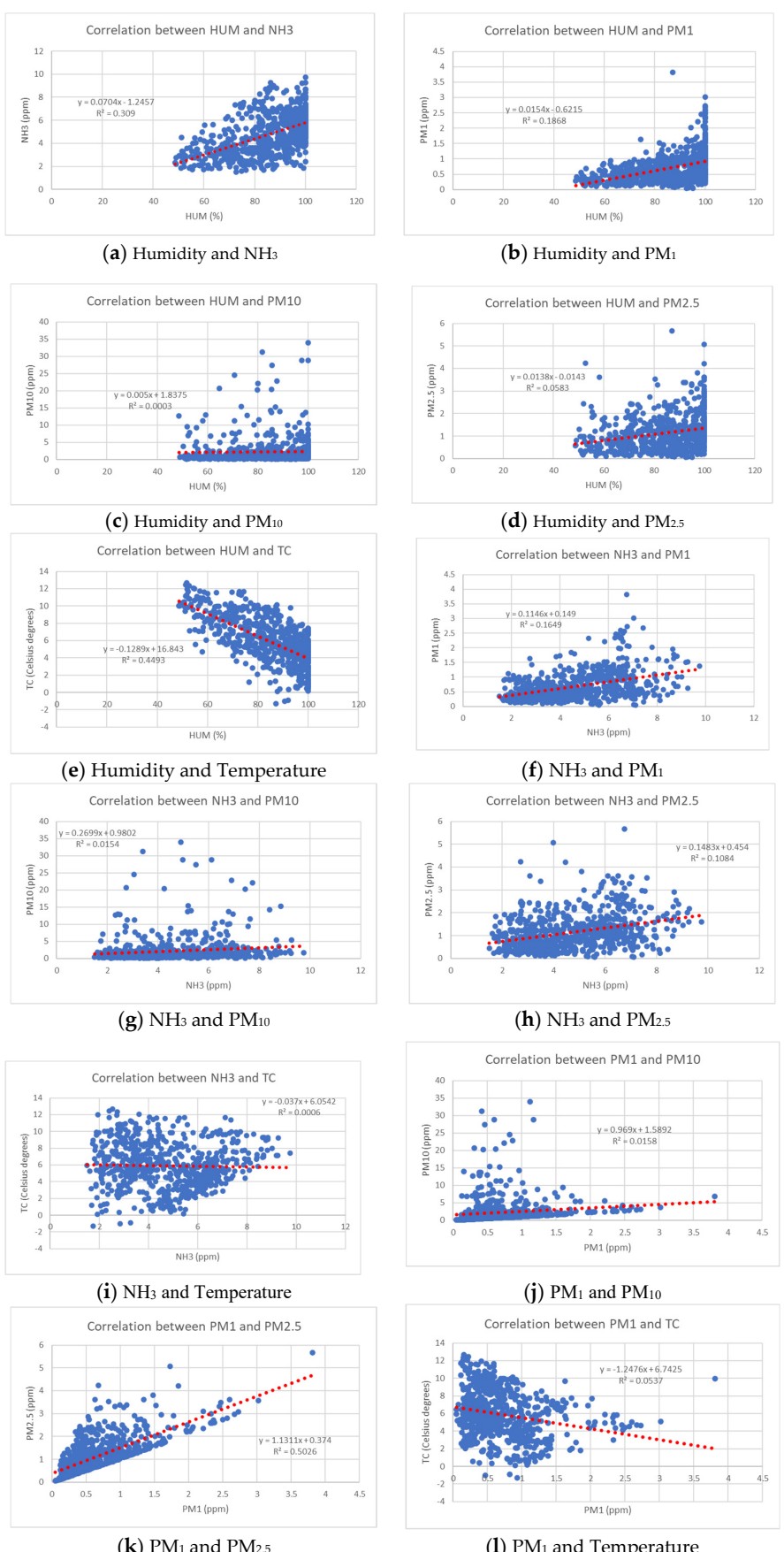

**Figure 3.** *Cont.*

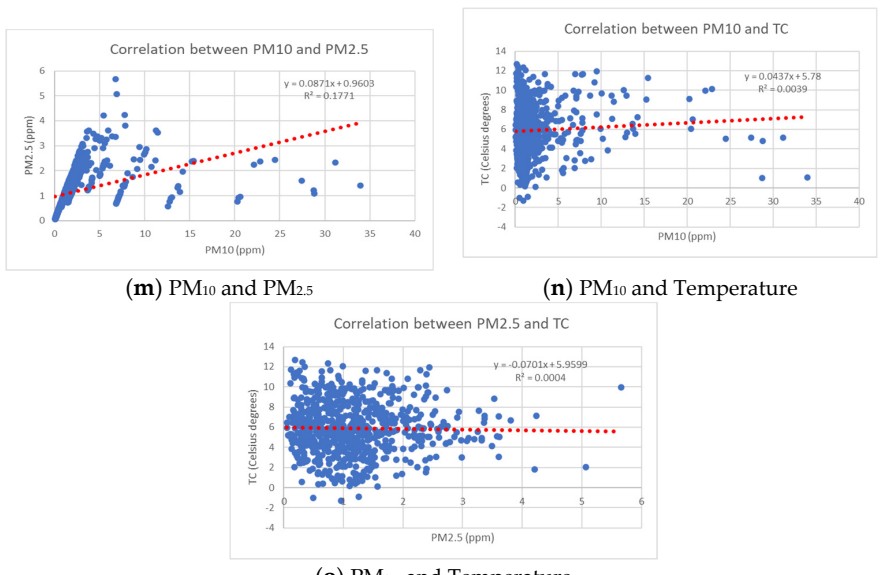

(**m**) PM$_{10}$ and PM$_{2.5}$   (**n**) PM$_{10}$ and Temperature

(**o**) PM$_{2.5}$ and Temperature

**Figure 3.** Relationship between different daily measured microclimate values and pollutants. The red line represents the 0−intercept line regression, during the winter season.

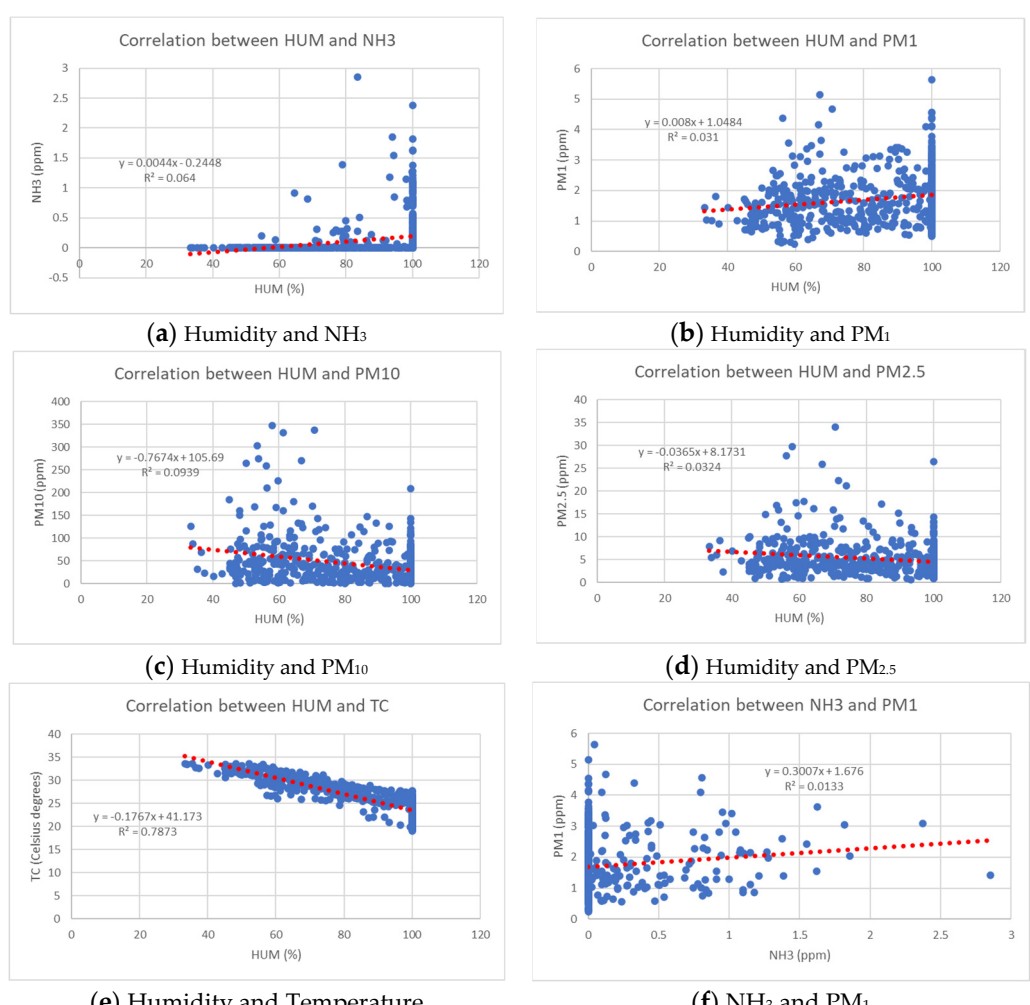

(**a**) Humidity and NH$_3$   (**b**) Humidity and PM$_1$

(**c**) Humidity and PM$_{10}$   (**d**) Humidity and PM$_{2.5}$

(**e**) Humidity and Temperature   (**f**) NH$_3$ and PM$_1$

**Figure 4.** *Cont*.

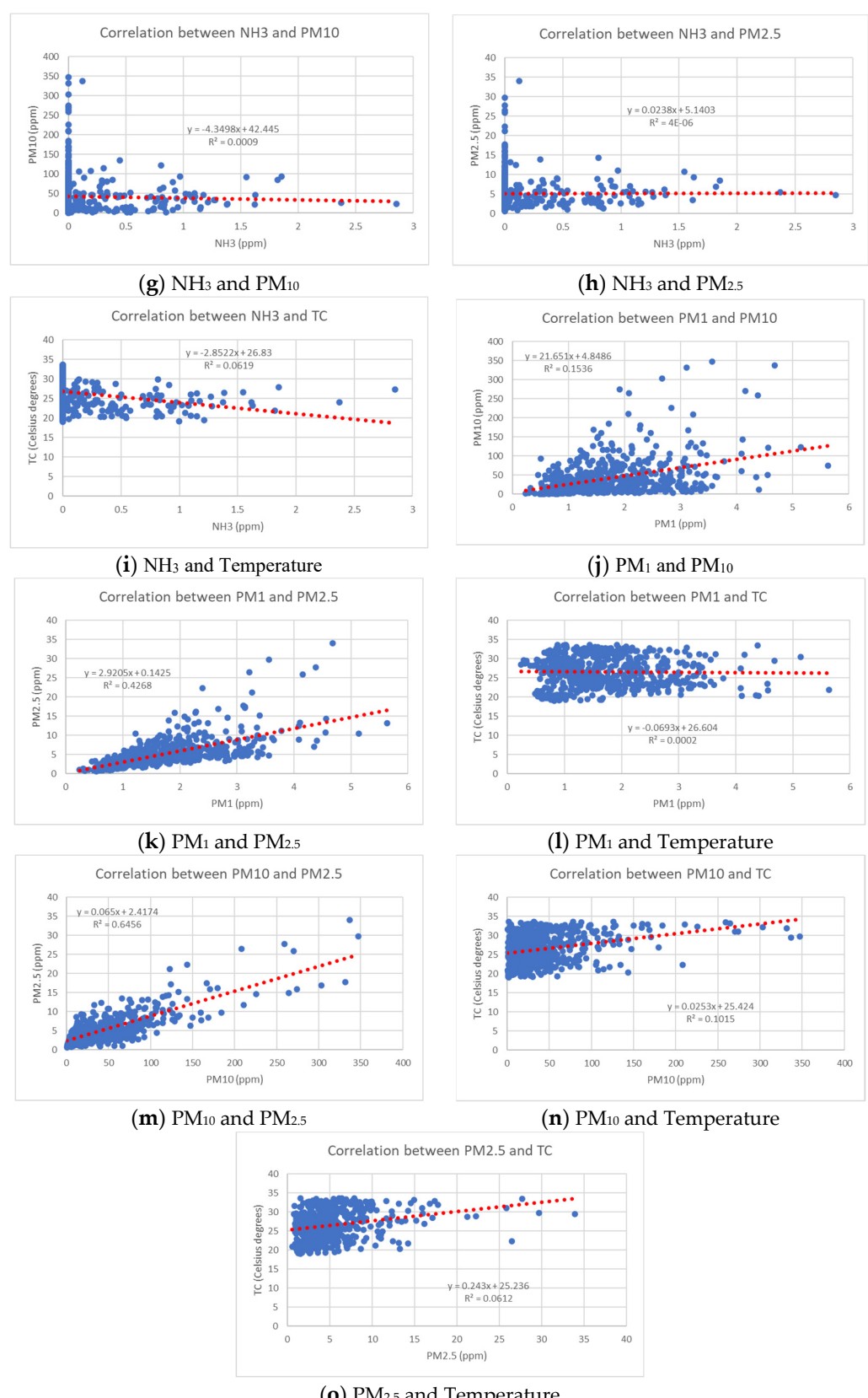

(**g**) NH₃ and PM₁₀

(**h**) NH₃ and PM₂.₅

(**i**) NH₃ and Temperature

(**j**) PM₁ and PM₁₀

(**k**) PM₁ and PM₂.₅

(**l**) PM₁ and Temperature

(**m**) PM₁₀ and PM₂.₅

(**n**) PM₁₀ and Temperature

(**o**) PM₂.₅ and Temperature

**Figure 4.** Relationship between different daily measured microclimate values and pollutants. The red line represents the 0-intercept line regression, during the summer season.

All these observations, which reveal a complexity in the behavior of microclimate parameters and pollutant emissions, complicate farm management activities, and the existence of a platform that allows permanent monitoring of air quality through sensors, as well as an alert system, becomes useful in interpreting the causality of concrete situations that go beyond the limits of admissibility and in the optimization of the decision-making system regarding the welfare of animals, workers and the environment.

Figure 5a–f presents the microclimate values and pollutant concentrations in the stable, throughout one day, in the two analyzed seasons. Measurements were made by placing sensors approximately 1 m above the animals' heads and approximately 2.5 m from the floor.

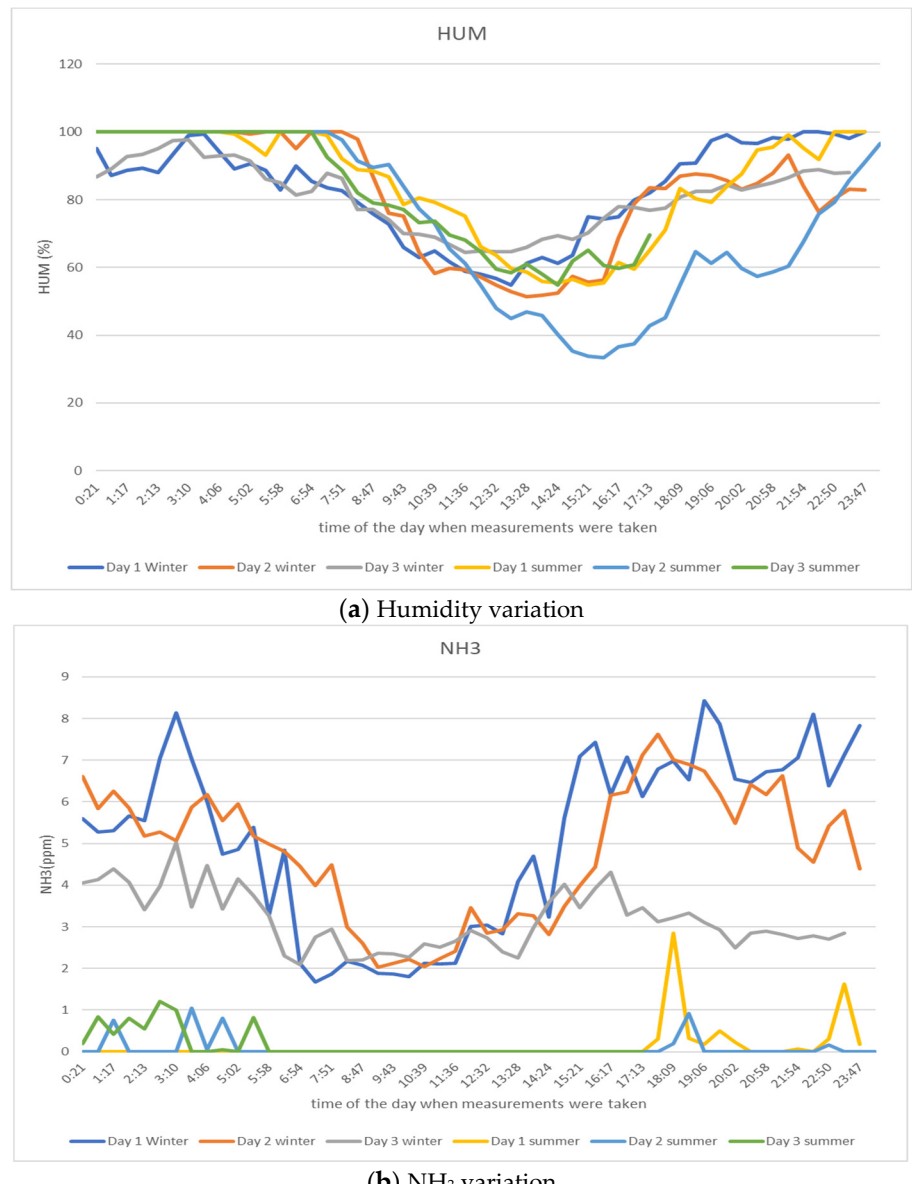

(**a**) Humidity variation

(**b**) $NH_3$ variation

**Figure 5.** *Cont.*

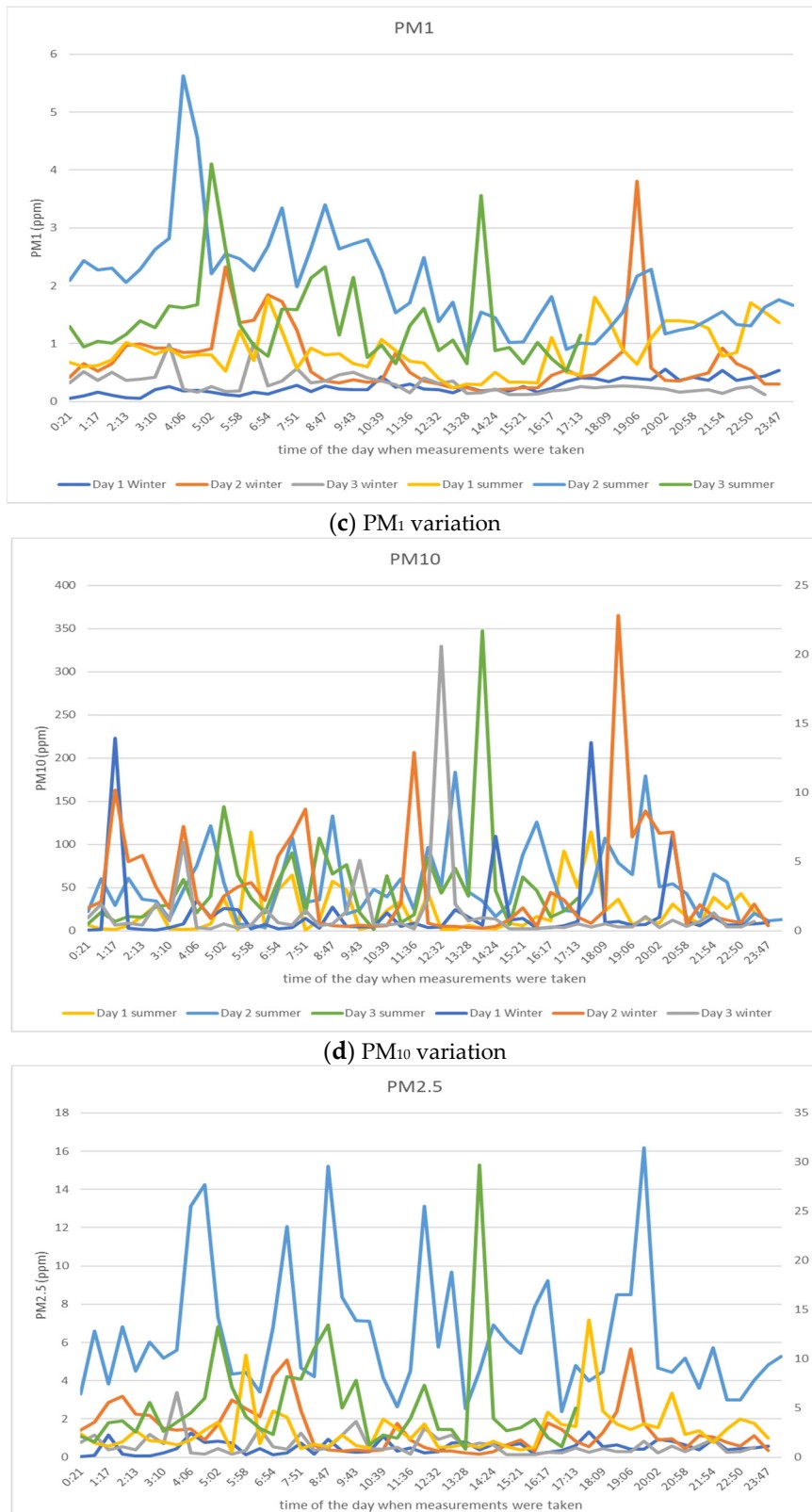

(**c**) PM$_1$ variation

(**d**) PM$_{10}$ variation

(**e**) PM$_{2.5}$ variation

**Figure 5.** *Cont.*

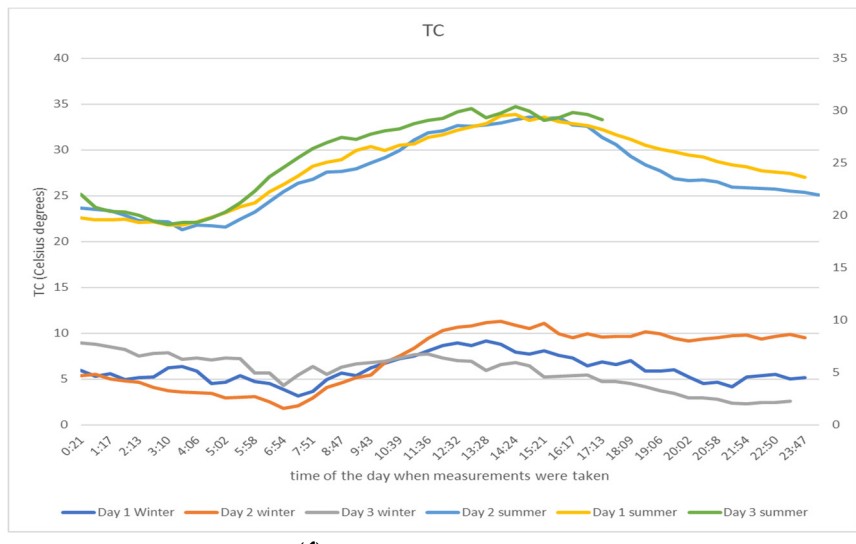

(**f**) Temperature variation

**Figure 5.** Microclimate values and pollutant concentrations at 1 m above the animals' heads in the barn, during one day, in both seasons: winter (January) and summer (August).

We can observe that, during the day, the concentration of ammonia increases when the stables are closed (during the night and afternoon to evening), a consequence of the lack of ventilation and the increase in humidity. Although the increase in ammonia concentration is apparently not related to temperature, it favors the increase in ammonia emissions by maintaining metabolic processes and intensifying urease activity in manure. Zero values for ammonia concentration during the warm season (no records) can be explained by its negative correlation with temperature, when the sensor was not able to record any emission (which does not mean that it did not exist, however, but simply that the values were below the sensitivity of the sensor).

The variation in $PM_{10}$ concentration captures the times of the day when food is administered and stable cleaning is performed. During both seasons, the concentration of $PM_{10}$ is also influenced by the increase in ammonia.

$PM_{2.5}$ concentrations throughout the day vary in the warm season following a similar pattern to $PM_{10}$, capturing the same activities, but in the cold season, throughout the day, $PM_{2.5}$ concentration variations most likely capture certain activities that are not are related to animal husbandry (possibly smoking in the barn by the farm hands, sheltered from the cold temperatures outside—this would also explain the $PM_1$ variations).

Regarding the observed difference between the values estimated by the EMEP/EEA guideline equations (2019) and the values recorded by the sensors, this phenomenon is because the estimates do not consider the natural and artificial ventilation of the stable, being used instead for an overall assessment of atmospheric air quality. Estimates and measurements may be similar only under experimental conditions, or if the stable were permanently closed. For this reason, for point sources of air pollutants from livestock (farms) it is recommended that farmers use sensors and create alert systems that adjust pollutant concentrations in real time (support for farm management decisions) to ensure animal welfare.

## 5. Conclusions

This article presents a case study of air pollutant emissions and their correlation with animal welfare and farm management in two different seasons.

Our use-case from the Milanovici farm showed that the estimated air pollutant concentration exhibits complex behavior that correlates with the micro-climate parameters. Therefore, for efficient farm management it is important to treat them as a whole and not individually.

We conclude that estimates using the EMEP methodology do not take into account the natural and artificial ventilation of the stable since they are used for an overall assessment of atmospheric air quality, and it is recommended that sensors are used and alert systems that adjust pollutant concentrations in real time (support for farm management decisions) are created to ensure animal welfare for point sources of air pollutants from livestock farms.

**Author Contributions:** Conceptualization, R.A.P., A.V., D.C.P. and L.V.; methodology, R.A.P., D.C.P., A.V., G.S. and M.P.M.; software, M.B., A.V., R.B. and S.B.; validation, R.A.P., D.C.P., A.V., M.T. and E.N.P.; writing—original draft preparation, R.A.P., D.C.P., M.B. and S.B.; writing—review and editing, A.V., D.C.P. and M.B.; project administration, A.V., D.C.P. and G.S.; funding acquisition, R.A.P. All authors have read and agreed to the published version of the manuscript.

**Funding:** This research was partially supported by a grant of the Romanian National Authority for Scientific Research and Innovation, CCCDI–UEFISCDI, projects no. ERANET-ERAGAS-ICTAGRI3-FarmSusteinaBl-1 and ERANET-ERAGAS-ICTAGRI3-FarmSusteinaBl-2 (FarmSustainaBl) within PNCDI III, and funded in part by a grant of Ministry of Research, Innovation and Digitization, CNCS/CCCDI-UEFISCDI, project number ERANET-ICT-AGRI-FOOD-Solution4Farming, within PNCD III. The authors acknowledge the financial support through the partners of the Joint Call of the Cofund ERA-Nets SusCrop (Grant N° 771134), FACCE ERA-GAS (Grant N° 696356), ICT-AGRI-FOOD (Grant N° 862665) and SusAn (Grant N° 696231) for the project Solution4Farming. The APC was funded by the University of Agricultural Sciences and Veterinary Medicine.

**Institutional Review Board Statement:** Not applicable.

**Data Availability Statement:** The data presented in this study are available on request from the corresponding authors.

**Acknowledgments:** The authors are grateful for the support of the Milanovici farm for making available their facilities for sensor installation and data collection and to Beia Cercetare SRL for providing the unified messaging and cloud data storage services.

**Conflicts of Interest:** The authors declare no conflict of interest. The funders had no role in the design of the study; in the collection, analyses, or interpretation of data; in the writing of the manuscript, or in the decision to publish the results.

## Abbreviations

In the manuscript, we used the following abbreviations and chemical symbols:

| | |
|---|---|
| AAP | Average Annual Population |
| ANFIS-GP | Adaptive Neuro-Fuzzy Inference Systems with Grid Partitioning |
| ANFIS-SC | Adaptive Neuro-Fuzzy Inference Systems with Subtractive Clustering |
| AP | Air Pollutant |
| AQI | Air Quality Index |
| AQM | Air Quality Monitoring |
| CFC | Cloud Farm Controller |
| CoAP | Constrained Application Protocol |
| CP | Crude Protein |
| EF | Emission Factor |
| EMEP | European Monitoring and Evaluation Programme |
| EPA | United States Environmental Protection Agency |
| EX-ACT | EX-Ante Carbon-balance Tool |
| FEM | Farm Emissions Model |
| HTTP | Hypertext Transfer Protocol |
| IoT | Internet of Things |
| IPCC | Intergovernmental Panel on Climate Change |
| KF | Kalman Filter |
| KPI | Key Performance Indicator |

| | |
|---|---|
| LEACH | Low Energy Adaptive Clustering Hierarchy Aggregation |
| LFC | Local Farm Controller |
| LMC | Litter Moisture Content |
| MLP | Multilayer Perceptron |
| MLR | Multiple Linear Regression |
| MQTT | Message Queuing Telemetry Transport |
| NPM | National Practices Model |
| pH | Potential of Hydrogen |
| PM | Microscopic Particles |
| PMx | Microscopic Particles less than x microns in diameter, where x {1, 2.5, 10} |
| REST | Representational State Transfer |
| WSN | Wireless Sensor Network |
| CO | carbon monoxide |
| $CO_2$ | carbon dioxide |
| $CH_4$ | methane |
| $N_2O$ | nitrous oxide |
| $NO_2$ | nitrogen dioxide |
| $NH_3$ | ammonia |
| $O_3$ | ozone |
| $SO_2$ | sulfur dioxide |

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
