# Peer review of "Comparative Evaluation of the Dynamics of Animal Husbandry Air Pollutant Emissions Using an IoT Platform for Farms"

_agriculture, doi:10.3390/agriculture13010025_

Round 1

Reviewer 1 Report

 The work is very interesting and the paper is written quite well. 

The following are the few suggestions to improv the paper

1. Over all network established for collection of data is not clear

2.one day data collected and analyzed in fig 5 may not be appropriate to draw conclusions.. the graphs must be drawn by collecting data over multiple iterations

3.what is the criteria used for selecting the periodicity is not clear?

4. Detailed analysis of the variation in pollutants and their effects has to be discussed

5. Proper inferences are to be drawn using all the correlation plots of Fig 3 and 4.

Reviewer 2 Report

1. EMEP was not define on first use in the Abstract

2. Both Micro-climate and microclimate were used, please be consistent with one of them.

3. The last statement of the abstract is quite vague, 

"they indicate possible activities done within the farm premises"

4. "at scale global", how about "at a global scale"?

5. Eutrophication seems more related to SDG 14 and 15, rather than 17.

You might want to have a look at this

5b. Line 52 and 53,  "...the United Nations have included this phenomenon in the list of objectives "Sustainable Development Goals (SDGs) 17""

How about "...the United Nations HAS included this phenomenon as part of the objectives of the "Sustainable Development Goals (SDGs), specifically objectives 14 and 15"

6. Please what is FAO as used on line 57? Food and Agriculture Organization (FAO)? 

7. Are lines 65-67 necessary to be a paragraph on their own? Why not combine them with the preceding paragraph, which discusses something related?

8. Line 78... "generates such pollutants [AS or AND] it is estimated..."

9. The inclusion of a reference/citation of the Eltonian pyramid might be useful on line 108, i.e., "... that the  Eltonian pyramid [X] is a .."

10. There are several paragraphs having just 1 or 2 statements, such as online line 164 - 166, 168 - 171, 172 - 176, 180-182. Why not combine paragraphs that are discussing related concepts into one.

11. The review of literature focused on the source of the pollutants. The authors did not really review  "related works" on IoT solutions in regards to farm pollution emission or similar

12. Please what is AP as used in lines 204 and 206, similarly what does EEA mean? Kindly define acroynms on their first use.

13. Line 208, based on these an IoT platform was designed... Only method 2 seems related to IoT, or are the authors implying that the EMEP/EEA methodology incorporates IoT methods?

14. Line 215, how about changing the caption of Fig. 1 to something like "IoT-based AP monitoring platform architecture"?

15. In Fig. 1, MQTT broker is shown, so I assume the authors are using a publish-subscribe model, how about adding 2 extra labels to the diagram, which indiate the  Publisher and subscriber respectively.

16. The statements on lines 231 - 234 are very debatable, for instance you included 4G, WiFi, Gateway, MQTT broker in Fig. 1. These devices do not generally run on batteries, and in cases where they do, the batteries do not last for 10 years+. The authors might want to specify that this statement refers mostly to the sensing layer.

17. Please what is a technological trailer?

18. Line 243, "...listed ON Table 1"

19. Line 245-247, " In Figure 2, it is seen the careful

placement of the IoT devices such that the everyday activities of the animals are not disturbed as well as the work done for configuring the devices"  

How about 

" Figure 2 shows that the IoT devices are carefully positioned in locations where they do not interfere with the daily activities of the animals...

19b. Please move the caption of Fig. 2 to the same page as the figure.

20. Could you please cite BME280 and OPC-N2. 

20b. CO, NO2, CO2, O2, VOC, are not Sensor names, please review these.

 21. Line 258, please change "This chapter" to "This section" 

22. Please cite IPCC 2019.

23. Please adjust the font size of Table 4, especially "kg/head/day"

24. It would have been nice if the authors included an exact definition of microclimate as used in the article.

25. On tables 9 and 10, t means correlation significance and also temperature. Yes, I can see that the temp has a degree symbol after it but it could also be misinterpreted to mean the degree of correlation significance instead of temperature in degrees celcius. The authors might want to use a completely different symbol to represent one of them. 

26. Lines 479 and 486, "in Figures 3 and 4"

27. In Figures 3c, 3f, 3g, and especially 3i and 3o, the correlations are not very clear and there seems to be many outliers that do not conform to this correlationship. The authors might want to elaborate more on the relations in Figures 31, 3o and 4b in particular. 

28. Line 501, Figures 5a to 5f present..

29. For the PM measurements, two observations come to mind, 

i. looking at Figure 1, it seems the farm stable was not 100% enclosed, hence, external air particles could have been measured

ii. The placement of the sensor (2.5m) from the ground (line 503) might suggest that the sensor picking more readings from external air blowing into the farm stable than within the stable itself. 

These factors might affect the accuracy of the measurements/readings.

30. Please have a look at Ref. 16 (line 598), remove the dash in protein. Similarly, Ref. 36 and 37 seem to be in a different referencing style than the others.
